# N2GON: Neural Networks for Graph-of-Net with Position Awareness

**Yejiang Wang** [1,2]  **Yuhai Zhao** [1,2]  **Zhengkui Wang** [3]  **Wen Shan** [4]  **Ling Li** [5]  **Qian Li** [6]  **Miaomiao Huang** [1,2]  **Meixia Wang** [1,2]  **Shirui Pan** [7]  **Xingwei Wang** [1]

## Abstract

Graphs, fundamental in modeling various research subjects such as computing networks, consist of nodes linked by edges. However, they typically function as components within larger structures in real-world scenarios, such as in protein-protein interactions where each protein is a graph in a larger network. This study delves into the Graph-of-Net (GON), a structure that extends the concept of traditional graphs by representing each node as a graph itself. It provides a multi-level perspective on the relationships between objects, encapsulating both the detailed structure of individual nodes and the broader network of dependencies. To learn node representations within the GON, we propose a position-aware neural network for Graph-of-Net which processes both intra-graph and inter-graph connections and incorporates additional data like node labels. Our model employs dual encoders and graph constructors to build and refine a constraint network, where nodes are adaptively arranged based on their positions, as determined by the network's constraint system. Our model demonstrates significant improvements over baselines in empirical evaluations on various datasets.

## 1. Introduction

Graphs, as mathematical structures that represent relationships between entities, pervade diverse domains. A conventional graph is a fundamental construct in graph theory,

[1]School of Computer Science and Engineering, Northeastern University, China [2]Key Laboratory of Intelligent Computing in Medical Image of Ministry of Education, Northeastern University, China [3]InfoComm Technology Cluster, Singapore Institute of Technology, SIT X NVIDIA AI Centre, Singapore [4]Singapore University of Social Sciences, Singapore [5]Shanxi University, China [6]Shandong University, China [7]Griffith University, Australia. Correspondence to: Yuhai Zhao <zhaoyuhai@mail.neu.edu.cn>.

*Proceedings of the 42nd International Conference on Machine Learning*, Vancouver, Canada. PMLR 267, 2025. Copyright 2025 by the author(s).

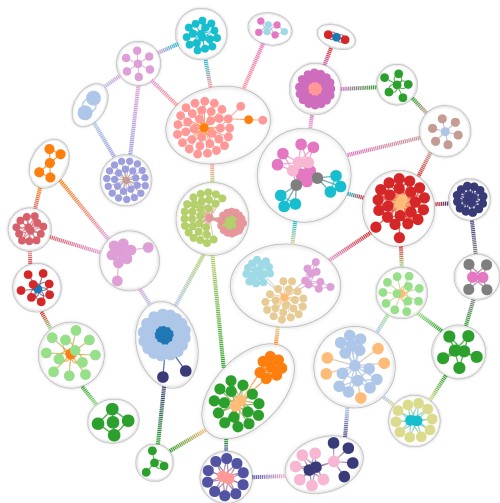

Figure 1: An example of a Graph-of-Net, where each node is a *graph* in itself, interconnected to form a *net*.

comprising a set of nodes and a set of edges, with each edge connecting two nodes. For example, in computing networks, nodes can represent computational units (such as servers, CPUs), while edges denote communication links between them. To address the intricacies of real-world applications, more intricate graph structures such as hypergraphs and heterogeneous graphs (Zhou et al., 2006; Zhang et al., 2019) have been employed. Hypergraphs extend the notion of standard graphs by allowing edges to connect any number of nodes, while heterogeneous graphs encompass various types of nodes and/or edges. Recent years have witnessed a burgeoning interest in the utilization of graph-structured data, including graph kernels (Tang & Yan, 2022), graph embeddings (Zhou et al., 2023), and graph neural networks (Liu et al., 2023; Wang et al., 2024a).

Despite substantial exploration in mathematics and various scientific fields, it becomes apparent that existing graph structures may inadequately capture the nuances of modern, complex real-world data, thereby hindering effective graph-based learning. Real-world scenarios exhibit a spectrum of graph structures that encompass diverse scales, types, and hierarchical levels. A myopic focus on individual graphs may overlook the intricate multi-level structures and inter-

dependencies inherent in these datasets.

To address this gap and effectively model a majority of real-world systems, this paper investigates a new graph structure, named **Graph-of-Net** (**GON**). Within a Graph-of-Net, each node itself is a graph. This construct incorporates two types of links: 1) intra-graph connections within each node-graph, and 2) inter-graph dependency links between node-graphs. Such a framework facilitates in-depth analysis and permits the exploration of relationships spanning multiple layers. An example, illustrated in Fig.1, arises in biological networks, where individual proteins are represented as small-scale graphs, while protein-protein interaction networks constitute large-scale graphs, often referred to as networks.

Existing graph learning methodologies typically either utilize multiple small-scale graphs to independently represent proteins, overlooking critical interactions, or amalgamate all proteins into a single large-scale graph, representing nodes as vectors, thus neglecting the intricate internal structure of proteins. GON shares conceptual similarities with previously used data structures in specific domains, such as the exploration of theoretical properties in physical sciences (Gao et al., 2011; Ni et al., 2014) or the prediction of interactions among drug molecules in the field of biology (Wang et al., 2021). However, it exhibits versatility across diverse domains. For instance, an Internet can be modeled as a GON, where local area networks interconnect as nodes within a vast web of data exchange. Similarly, a citation network can be captured as a GON, wherein individual papers, conceptualized as text graphs, interconnect through citations.

This paper delves into the intricacies of GON structures, their representation, and their applicability in modeling complex real-world systems. By embracing the inherent graph-of-net paradigm, we aim to provide a comprehensive framework that addresses the multifaceted nature of modern data structures and their interdependencies.

The Graph-of-Net (GON) presents a paradigm divergent from conventional graphs. As such, traditional graph learning methodologies, such as graph neural networks (GNNs), are not inherently suited for direct application to GONs. To efficaciously learn the representations within GONs, we introduce an position-aware neural network specifically tailored for Graph-of-Net (N2GON). This model meticulously incorporates the dual link types inherent in GONs while synergistically integrating supplementary data, such as node labels, to discern and encapsulate more nuanced patterns and structural complexities. Our approach involves the development of dual graph encoders: one dedicated to embedding the constituent node-graphs and the other to embedding the overarching network, utilizing the representations derived from the first encoder. Furthermore, we construct an implicit constraint network that interconnects these graph nodes based on their labels and contextual relationships with positions. This network is conceived from two critical insights: firstly, nodes with similar labels should have closely aligned embeddings, whereas those with dissimilar labels should diverge; secondly, the interactive proximity between node-graphs diminishes as the separation between their relative positions grows. By employing this constraint network, we refine the similarities among the node embeddings. This layered, network-wide approach, which accounts for both intra-graph and inter-graph dynamics provides a framework for understanding the complex structures. Our contributions are summarized as follows:

- We explore a novel Graph-of-Net structure that lends itself to easy generalization across various fields, e.g., citation networks, and biomedical networks.

- We propose a position-aware neural network designed specifically for the GON structure. Our model innovatively manages both intra- and inter-graph interactions, significantly enhancing our ability to decipher and interpret intricate data.

- We conduct extensive experiments on various types of data, including 9 benchmark network datasets and 7 biomedical datasets. The results show that our model significantly outperforms SOTA baselines.

## 2. Related Work

### 2.1. Graph and Graph-of-Net

Graphs, with vertices and edges, serve well for depicting binary relationships (West et al., 2001; Wang et al., 2024b;c). The discipline has evolved, incorporating complex structures such as hypergraphs (Zhou et al., 2006) and heterogeneous graphs (Liu et al., 2024a; Mo et al., 2024). Hypergraphs permit edges to connect multiple vertices, enabling the representation of complex many-to-many relationships. Heterogeneous graphs, characterized by their diverse types of vertices and edges. In this evolving landscape, the concept of the GON introduces a distinct conceptual framework. In contrast to these traditional graphs, the GON incorporates complete graphs within the nodes of a larger graph. This approach offers a unique perspective for representing hierarchical and multi-layered relationships. However, existing research on Graph-of-Net is primarily domain-specific. For instance, in the physical sciences, (Gao et al., 2011) has been dedicated to establishing a precise percolation law applicable to a network comprising $n$ interdependent networks. Similarly, in the pharmaceutical sector, research (Wang et al., 2021) has predominantly focused on the prediction of drug-drug/chemical-chemical interactions. In contrast to these specialized domains, our study adopts a broader approach. We concentrate on universal structure

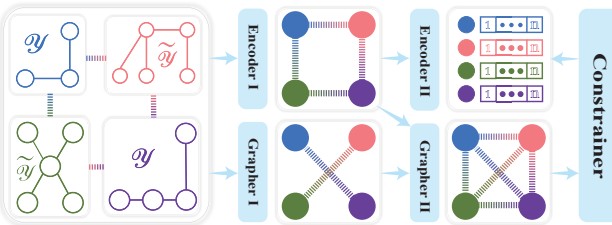

Figure 2: The framework of the N2GON model.

of GON, aiming for applicability in a wide range of fields. This includes, but is not limited to, social networks, citation networks, and biomedical networks.

## 2.2. Graph Representation Learning

Graph representation learning (GRL) is experiencing significant development, focusing on transforming input nodes into low-dimensio- nal vectors. These vectors are primarily used for tasks at both the graph and node levels. The initial phase of GRL centered around shallow networks, with notable examples including DeepWalk (Perozzi et al., 2014), which employed random walks to generate node embeddings, and Node2Vec (Grover & Leskovec, 2016), which introduced a more flexible approach for neighborhood sampling. In recent years, the rise of graph neural networks (GNNs) has marked a significant advancement in graph-based learning. GNNs operate by aggregating the representations of both a node and its neighbors in a recursive manner. They achieved state-of-the-art performance on various tasks (Liu et al., 2024b), such as node classification (Zhang et al., 2024; Wang et al., 2023), node clustering (Tsitsulin et al., 2023), etc. Nevertheless, despite these significant strides, applying these advanced techniques to GON structures, known for their hierarchical and multi-relational complexity, remains a challenging and active area of research. This highlights the need for exploration in the domain of GON learning.

## 2.3. Graph Coasrening Learning

Graph coarsening, a technique devised to reduce the complexity of large graphs, involves the consolidation of multiple nodes into a single node. This process traditionally relies on mathematical methods, which is exemplified in (Dhillon et al., 2007) through graph cuts. Recent advancements, like those by (Cai et al., 2021), have incorporated graph neural networks to refine edge weights, enhancing coarsening efficiency. Additionally, (Jin et al., 2022) have contributed by focusing on preprocessing and graph condensation, crucial for the scalability and acceleration of graph meural networks. Graph coarsening and GON differ significantly in their functions. Graph coarsening simplifies a single graph for better processing efficiency, while GON, a complex structure, represents multi-layered systems with

each node as a graph, highlighting data's hierarchical and interconnected aspects.

## 3. Methodology

In this section, we will introduce a structure, named Graph-of-Net (GON), and propose a novel neural network model for learning from Graph-of-Net data. Actually, the Graph-of-Net structure is a generalization of the traditional graph structure, where each node is a graph. Given an input Graph-of-Net, we aim to learn the representations of nodes in it. To effectively achieve this goal, we propose an position-aware neural network model, which considers both intra-graph connections within node-graphs and inter-graph dependencies among them and integrates additional data like node labels to capture complex patterns.

The framework of the proposed model is illustrated in Fig.2. We employ two encoders, *Encoder I* and *Encoder II*, to encode the Graph-of-Net, with the former targeting the node-graphs and the latter focusing on the Net formed between these node-graphs. Two graph constructors, *Grapher I* and *Grapher II*, are utilized to build a node-graph constraint network. *Grapher I*, based on the labels of the node-graphs, connects two nodes with the same label with an edge. *Grapher II*, leveraging this preliminary network in conjunction with the Graph-of-Net's inherent structure, computes the final constraint network, which features edge weights indicating the similarity between nodes. Guided by this network, the *Constrainer* strategically arranges the node-graph representations based on their *relative positions*, aligning those with greater similarities in closer proximity to each other, while distancing those is dissimilar (i.e., with different labels).

### 3.1. Definition of Graph-of-Net

A Graph-of-Net is mathematically characterized as a sophisticated structure $\mathcal{G}^{\mathsf{N}} = (\mathcal{V}_{\mathcal{G}}, \mathcal{E}_{\mathcal{G}})$, wherein the node set $\mathcal{V}_{\mathcal{G}}$ comprises a collection of $N$ individual graphs, each paired with its corresponding one-hot label, represented as $\{(\mathsf{G}_i, \mathsf{Y}_i) | i \in \{1, \ldots, N\}\}$. The edge set $\mathcal{E}_{\mathcal{G}} \subseteq \{(\mathsf{G}_i, \mathsf{G}_j) | \mathsf{G}_i, \mathsf{G}_j \in \mathcal{V}_{\mathcal{G}}\}$ delineates the inter-graph relationships within this structure. The adjacency matrix $\mathcal{A}_{\mathcal{G}}$ for $\mathcal{G}^{\mathsf{N}}$ is defined such that each element $\mathcal{A}_{ij}$ signifies the connectivity between graphs $\mathsf{G}_i$ and $\mathsf{G}_j$, encapsulating the comprehensive network of interactions. Each graph node $\mathsf{G}_i$ is defined as $\mathsf{G}_i = (\mathsf{V}_i, \mathsf{E}_i, \mathsf{X}_i)$, where $\mathsf{V}_i = \{\mathsf{v}_1^i, \ldots, \mathsf{v}_n^i\}$ and $\mathsf{E}_i \subseteq \{(\mathsf{v}_s^i, \mathsf{v}_t^i) | \mathsf{v}_s^i, \mathsf{v}_t^i \in \mathsf{V}_i\}$ represent the vertices and edges of $\mathsf{G}_i$, respectively. It's important to note that $\mathsf{E}_i$ symbolizes the intra-graph connections. The feature matrix $\mathsf{X}_i \in \mathbb{R}^{n \times d}$ encapsulates the attributes of vertices in $\mathsf{G}_i$, with $n$ being the number of vertices and $d$ the dimensionality of these features. The adjacency matrix $\mathsf{A}_i$ of $\mathsf{G}_i$ assigns a value of 1 to $\mathsf{A}_{ij}$ if $(\mathsf{v}_s^i, \mathsf{v}_t^i) \in \mathsf{E}_i$ and 0 otherwise.

In this work, given a Graph-of-Net $\mathcal{G}^{\mathsf{N}}$, our primary objective is to engage in the task of Graph-of-Net supervised learning. This involves learning the representative vectors $\mathsf{v_G}$ for the graph nodes, which are instrumental in predicting their labels.

## 3.2. Encoders of Graph-of-Net

Given the intricate composition of a Graph-of-Net, where each node themselves possesses distinct structures and the inter-node relationships coalesce into a network, we adopt a dual-encoder strategy. This approach is tailored to harness the multifaceted structural nuances of the Graph-of-Net: 1) *Encoder I*, tasked with transforming individual node-graphs into low-dimensional representations; 2) *Encoder II*, which, building upon the foundations laid by the *Encoder I*, further refines these representations in alignment with the GON's inherent network architecture. This sequential, dual-encoder setup is meticulously designed to integrate the two-tiered structural essence of the Graph-of-Net, thus ensuring that the resultant node representations are a comprehensive blend of both the nodes' intrinsic structural details and the overarching inter-node relational dynamics.

In the context of a specific Graph-of-Net $\mathcal{G}^{\mathsf{N}}$, we employs a message-passing framework to encode the node-graph $\mathsf{G}$, with a focus on retaining the vital adjacency information intrinsic to the node-graph as follows. Let $h_s^\ell \in \mathbb{F}_\ell$ denotes the feature at layer $\ell$ associated with node $s$, where $\mathbb{F}$ denotes arbitrary finite-dimensional space of the form $\mathbb{R}^q$ (for various values of $q$) typically representing the feature space, the updated feature $h_s^{\ell+1}$ is obtained as: $h_s^{\ell+1} = f_{upd}(h_s^\ell, \{\{h_t^\ell | t \in \mathcal{N}_s\}\})$, where $t \in \mathcal{N}_s$ means that nodes $t$ and $s$ are neighbors in the graph $\mathsf{G}$, i.e. $(s,t) \in \mathsf{E}$, and the function $f_{upd} : 2^{\mathbb{F}_\ell} \to \mathbb{F}_{\ell+1}$ is a learnable function taking as input the feature vector of the center vertex $h_s^\ell$ and the multiset of features of the neighboring vertices $\{\{h_t^\ell | t \in \mathcal{N}_s\}\}$. Indeed, for any such function $f_{upd}$ can be approximated by a layer of the form

$$h_s^{\ell+1} = \sigma\left(W^\ell \cdot \left(h_s^\ell \otimes f^\ell\left(h_s^\ell, \{\{h_t^\ell | t \in \mathcal{N}_s\}\}\right)\right)\right) \quad (1)$$

where $f^\ell : 2^{\mathbb{F}_\ell} \to \mathbb{F}_{\ell+1}$ is injective set functions in the $\ell$-th layer, $\otimes$ denotes vector concatenation, $W^\ell$ is learnable weight matrix and $\sigma$ is an element-wise activation function. We get the $\ell$-th message passing layer $f_{\mathsf{G}}^\ell : \mathbb{F}_\ell \to \mathbb{F}_{\ell+1}$ (note that $f_{\mathsf{G}}$ depends implicitly on the node-graph). Then, by the composition of $f_{\mathsf{G}}^\ell$ and pool function $f_{\mathsf{G}}^{\text{Pool}} : \mathbb{F}_{L_1} \to \mathbb{F}_{L_1+1}$, we obtain the representation of each graph node

$$\mathsf{h_G} = f_{\mathsf{G}}^{\text{Pool}} \circ f_{\mathsf{G}}^{L_1} \circ \ldots f_{\mathsf{G}}^2 \circ f_{\mathsf{G}}^1(\mathsf{X}) \quad (2)$$

where $L_1$ denotes the number of layers used in the node-graph encoder. Leveraging the derived representations of node-graphs, the *Encoder II* adopts a message propagation

technique to further update these representations within the GON network. Analogously, we establish the $\ell$-th message passing layer $f_{\mathcal{G}}^\ell : \mathbb{F}_\ell \to \mathbb{F}_{\ell+1}$, with a consideration that $f_{\mathcal{G}}$ is network-dependent. Subsequently, by orchestrating a series of operations through $f_{\mathcal{G}}^\ell$, we update the representation of each node

$$\mathsf{v_G} = f_{\mathcal{G}}^{L_2} \circ \ldots \circ f_{\mathcal{G}}^1(\mathsf{h_G}) \quad (3)$$

in the GON network, where $L_2$ is the number of layers used in the net encoder.

## 3.3. Construction of Constraint Network

Upon finalizing the representations of node-graphs within the context of supervised learning—where each node-graph's label is accessible during training—we focus on constraining these representations using their associated labels. Given the node label matrix $\mathbf{Y} \in \mathbb{R}^{n \times q}$, where row $\mathsf{Y}_i$ denotes the one-hot label of the $i$-th node-graph, we define the label identity matrix $\mathbf{C}$ as

$$\mathbf{C} = \mathbf{Y} \times \mathbf{Y}^\top \quad (4)$$

to construct a constraint network (*Grapher I*). This matrix's elements, $\mathbf{C}_{ij}$, serve to denote whether the labels of the $i$-th and $j$-th node-graphs match, assigning a value of 1 for identical labels and 0 for differing ones.

A direct approach might suggest that pairs of nodes with 1s in matrix $\mathbf{C}$ are similar, whereas those corresponding to 0s are dissimilar. However, this method neglects the nuanced interrelations among node-graphs, treating nodes as if they were independently and identically distributed. It simplistically presumes uniform similarity for nodes sharing labels, assigning a consistent weight of 1.

Yet, in practical scenarios, two nodes sharing a label may exhibit varying levels of similarity. For example, in a citation network, a paper $P_1$ focused on graph neural networks (GNNs) and two other papers $P_2$ and $P_3$, although sharing a common thematic label (e.g., neural networks), may differ in their degree of relevance to $P_1$. If $P_2$ discusses GNNs while $P_3$ explores convolutional neural networks (CNNs), then $P_2$ inherently shares a closer similarity with $P_1$ than $P_3$. To tackle this, we recognize that the GON's inherent network structure provides critical insights to differentiate such varying degrees of similarity. Nodes sharing a label and exhibiting closer connections should logically possess higher similarity, and the converse for less connected nodes. In our example, paper $P_1$ is more likely to cite paper $P_2$ (directly interconnected) than $P_3$, suggesting a tighter connection within the GON's citation network.

Building upon these insights, we address the issue of uniform similarity weights by evaluating node significance and adjacency within the GON. To this end, we construct a refined constraint network (*Grapher II*), which recalibrates

---

**Algorithm 1** N2GON

---

**Input :** GON $\mathcal{G}^{\mathsf{N}}$. $\mathcal{F}_1, \mathcal{F}_2$: the backbone encoders.
**while** *not converge* **do**
    Sample full-batch of node-graphs $\{\mathsf{G}_i\}_{i=1}^{N}$ from $\mathcal{G}^{\mathsf{N}}$;
    Encode $\{\mathsf{G}_i\}_{i=1}^{N}$ by $\mathcal{F}_1$ Eq.(2) to get $\{\mathsf{h}_{\mathsf{G}_i}\}_{i=1}^{N}$;
    Update $\{\mathsf{h}_{\mathsf{G}_i}\}_{i=1}^{N}$ by $\mathcal{F}_2$ Eq.(3) to get $\{\mathsf{v}_{\mathsf{G}_i}\}_{i=1}^{N}$;
    Generate constraint net **C** Eq.(4) and $\mathbf{\Pi}$ Eq.(6);
    Calculate constrain loss $\mathcal{L}_{\text{con}}$ Eq.(7) and NLL loss, opti-
      mize the encoders $\mathcal{F}_1, \mathcal{F}_2$;
**end**
**Output :** The well trained model $\mathcal{F}_1, \mathcal{F}_2$

---

the similarity weights among node-graphs. This recalibration is inspired by the Personalized PageRank (PPR) algorithm. The PPR algorithm calculates the probability of an $\alpha$-random walk, which originates from a source node and concludes at a target node. In our model, the source node corresponds to the node-graph $\mathsf{G}_i$, and the target node is $\mathsf{G}_j$. Here, an $\alpha$-random walk represents a path taken by a random surfer on the network, who either halts at the current node with a probability of $\alpha$ or proceeds to a randomly selected outgoing neighbor with a probability of $1 - \alpha$. The pairwise score value $\pi(\mathsf{G}_i, \mathsf{G}_j)$ can be computed by solving the following equation:

$$\pi(\mathsf{G}_i, \mathsf{G}_j) = \alpha \mathbb{1}(\mathsf{G}_i, \mathsf{G}_j) + (1 - \alpha) \sum_{\mathsf{G}_k \in \mathcal{N}_{\mathsf{G}_j}} \frac{\pi(\mathsf{G}_i, \mathsf{G}_k)}{d(\mathsf{G}_k)} \quad (5)$$

wherein $\mathcal{N}_{\mathsf{G}}$ signifies the set of neighboring node-graphs of $\mathsf{G}$, $\mathbb{1}(\cdot, \cdot)$ is an indicator, and $d(\mathsf{G})$ denotes the degree of node-graph $\mathsf{G}$. Intuitively, as per this definition, the value $\pi(\mathsf{G}_i, \mathsf{G}_j)$ assesses the *importance* and *similarity* of a node $\mathsf{G}_j$ in relation to the source $\mathsf{G}_i$. Let $\mathbf{\Pi}$ represent the score matrix, where $(\mathbf{\Pi})_{ij}$ corresponds to the importance value $\pi(\mathsf{G}_i, \mathsf{G}_j)$ relative to the source $i$. We have

$$\mathbf{\Pi} = \alpha(\mathbf{I}_n - (1 - \alpha)\mathcal{A}_{\mathcal{G}})^{-1} \quad (6)$$

where $\mathbf{I}_n$ is the identity matrix. Hence, our model considers not only the label information of graph nodes but also the *positional* information between nodes, which computes the topological influence of node-graphs on each other, implicitly revealing their relative positions in the network.

### 3.4. Loss

After establishing the constraint network, we integrate it with the representations of our node-graphs. The goal here is to meticulously align these representations based on the pairwise importance scores defined within the constraint network. This strategic alignment is crafted to bring representations of node-graphs with higher similarity scores closer together, while placing those with lower similarity scores at a proportionately greater distance, and explicitly

segregating the representations of node-graphs that are dissimilar (namely, those with differing labels). To achieve this goal, for each node-graph, we define a loss function to constrain it in relation to other nodes (*Constrainer*):

$$\mathcal{L}_{\text{con}} = -\sum_{ij}^{N} (\mathbf{\Pi} \circ \mathbf{C})_{ij} \, \log \frac{\exp(sim(\mathsf{v}_{\mathsf{G}_i}, \mathsf{v}_{\mathsf{G}_j})/\tau)}{\sum_{k \neq i}^{N} \exp(sim(\mathsf{v}_{\mathsf{G}_i}, \mathsf{v}_{\mathsf{G}_k})/\tau)} \quad (7)$$

where $\circ$ denotes the Hadamard product, $\mathsf{v}_{\mathsf{G}}$ is the representation of node-graphs $\mathsf{G}$. $sim(\mathsf{a}, \mathsf{b})$ denotes the similarity between vectors $\mathsf{a}$ and $\mathsf{b}$, which is usually defined as the cosine similarity without normalization. $\tau$ is a temperature parameter. For label-based supervision, we employ the Negative Log-Likelihood (NLL) as the loss function. Consequently, the overall loss is formulated as the sum of the constraint loss and the NLL loss. The algorithm of the proposed model N2GON is summarized in Algorithm 1.

## 4. Computational Complexity

In this section, we provide a comprehensive discussion of the computational complexity of the proposed N2GON approach. Let the number of node-graphs in the Graph-of-Net be $|\mathcal{V}|$, and the number of edges between the node-graphs be $|\mathcal{E}|$. Additionally, the number of nodes and edges within node-graph are denoted as $|V|$ and $|E|$, respectively. 1) Encoder: The time complexity of the graph neural network backbone for Encoder I and II are $O(|V| + |E|)$ and $O(|\mathcal{V}| + |\mathcal{E}|)$, respectively. 2) Grapher I: Since the node label matrix $\mathbf{Y}$ is a sparse one-hot label matrix, the computation of the label identity matrix $\mathbf{C}$ is mainly related to the number of non-zero elements in $\mathbf{Y}$. Therefore, the time complexity is $O(\delta|\mathcal{V}|q)$, where $\delta << 1$ represents the ratio of non-zero elements in $\mathbf{Y}$, and $q$ is the number of classes. 3) Grapher II: In constructing the refined constraint network, we use the PageRank local partitioning algorithm, with a logarithmic running time of $O\left(\frac{2^b \log^3 |\mathcal{E}|}{\phi^2}\right)$, where $\phi$ is the conductivity and $b \in [1, \log m]$ is a constant. 4) Loss: Since the constraint loss requires the computation of the dot product between pairs of embeddings, the time complexity is $O\left(|\mathcal{V}|^2 d\right)$, where $d$ is the dimensionality of the embeddings. Therefore, the total computational time complexity is $O\left(\delta|\mathcal{V}|q + |\mathcal{E}| + \frac{2^b \log^3 |\mathcal{E}|}{\phi^2} + |\mathcal{V}|^2 D + |V| + |E|\right)$.

## 5. Experiments

In this section, we conduct an extensive experimental evaluation of N2GON, examining its performance across a wide range of datasets from various domains, including social networks, citation networks, web page networks, and biomedical data. The first three categories are common in graph learning research, allowing us to compare N2GON with state-of-the-art graph learning algorithms. However,

Table 1: Mean test classification accuracy (%) ± stdev on 6 heterophily and 3 homophily datasets. The highest performance is highlighted, and the second best performance is underlined.

| | Texas | Wisconsin | Actor | Squirrel | Chameleon | Cornell | Citeseer | Pubmed | Cora |
|---|---|---|---|---|---|---|---|---|---|
| Hom. level $h$ | 0.11 | 0.21 | 0.22 | 0.22 | 0.23 | 0.3 | 0.74 | 0.8 | 0.81 |
| #Nodes $|\mathcal{V}|$ | 183 | 251 | 7,600 | 5,201 | 2,277 | 183 | 3,327 | 19,717 | 2,708 |
| #Edges $|\mathcal{E}|$ | 295 | 466 | 26,752 | 198,493 | 31,421 | 280 | 4,676 | 44,327 | 5,278 |
| #Classes | 5 | 5 | 5 | 5 | 5 | 5 | 7 | 3 | 6 |
| MLP | 81.89±4.78 | 85.29±3.61 | 35.76±0.98 | 29.68±1.81 | 46.36±2.52 | 81.08±6.37 | 72.41±2.18 | 86.65±0.35 | 74.75±2.22 |
| GCN | 55.14±5.16 | 51.76±3.06 | 27.32±1.10 | 53.43±2.01 | 64.82±2.24 | 60.54±5.30 | 76.50±1.36 | 88.42±0.50 | 86.90±1.04 |
| GAT | 52.14±5.16 | 49.41±4.09 | 27.44±0.89 | 40.72±1.55 | 60.26±2.50 | 61.89±5.05 | 76.55±1.23 | 86.33±0.48 | 87.30±1.10 |
| GraphSAGE | 82.43±6.14 | 81.18±5.56 | 34.23±0.99 | 41.61±0.74 | 58.73±1.68 | 75.95±5.01 | 76.04±1.30 | 88.45±0.50 | 86.90±1.04 |
| GCNII | 77.57±3.83 | 80.39±3.40 | 37.44±1.30 | 38.47±1.58 | 63.86±3.04 | 77.86±3.79 | 77.33±1.48 | 90.15±0.43 | 88.37±1.25 |
| H2GCN-1 | 84.86±6.77 | 86.67±4.69 | 35.86±1.03 | 36.42±1.89 | 57.11±1.58 | 82.16±4.80 | 77.07±1.64 | 89.40±0.34 | 86.92±1.37 |
| H2GCN-2 | 82.16±5.28 | 85.88±4.22 | 35.62±1.30 | 37.90±2.02 | 59.39±1.98 | 82.16±6.00 | 76.88±1.77 | 89.59±0.33 | 87.81±1.35 |
| ACM-GCN | 87.84±4.40 | 88.43±3.22 | 36.28±1.09 | 54.40±1.88 | 66.93±1.85 | 85.14±6.07 | 77.32±1.70 | 90.00±0.52 | 87.91±0.95 |
| WRGAT | 83.62±5.50 | 86.98±3.78 | 36.53±0.77 | 48.85±0.78 | 65.24±0.87 | 81.62±3.90 | 76.81±1.89 | 88.52±0.92 | 87.95±1.18 |
| GGCN | 84.86±4.55 | 86.86±3.29 | 37.54±1.56 | 55.17±1.58 | 71.14±1.84 | 85.68±6.63 | 77.14±1.45 | 89.15±0.37 | 87.95±1.05 |
| S2GC | 68.65±8.05 | 71.57±9.01 | 34.17±0.92 | 41.63±0.98 | 58.55±5.15 | 75.25±7.82 | 76.08±0.45 | 88.31±0.38 | 87.73±2.90 |
| SIGN | 75.14±7.94 | 80.59±3.75 | 36.14±1.01 | 40.16±2.12 | 60.48±2.10 | 78.11±4.67 | 76.53±1.76 | 89.58±0.45 | 86.72±1.37 |
| APPNP | 78.37±6.01 | 81.42±4.34 | 34.64±1.51 | 33.51±2.02 | 47.50±1.76 | 77.02±7.01 | 77.06±1.73 | 87.94±0.56 | 87.71±1.34 |
| GPRGNN | 82.12±7.72 | 81.16±3.17 | 33.29±1.39 | 43.29±1.66 | 61.82±2.39 | 81.08±6.59 | 75.56±1.62 | 86.85±0.46 | 86.98±1.33 |
| GCN+JK | 66.49±6.64 | 74.31±6.43 | 34.18±0.85 | 40.45±1.61 | 63.42±2.00 | 64.59±8.68 | 74.51±1.75 | 88.41±0.45 | 86.79±0.92 |
| GCN-Cheby | 77.30±4.07 | 79.41±4.46 | 34.11±1.09 | 43.86±1.64 | 55.24±2.76 | 74.32±7.46 | 75.82±1.53 | 88.72±0.55 | 86.76±0.95 |
| MixHop | 77.84±7.73 | 75.88±4.90 | 32.22±2.34 | 43.80±1.48 | 60.50±2.53 | 73.51±6.34 | 76.26±1.33 | 85.31±0.61 | 87.61±0.85 |
| FAGCN | 78.11±5.01 | 81.56±4.64 | 35.41±1.18 | 42.43±2.11 | 56.31±3.22 | 76.12±7.65 | 74.86±2.42 | 85.74±0.36 | 83.21±2.04 |
| DAGNN | 70.27±4.93 | 71.76±5.25 | 35.51±1.10 | 30.29±2.23 | 45.92±2.30 | 73.51±7.18 | 76.44±1.97 | 89.37±0.52 | 86.82±1.67 |
| HopGNN | 82.97±5.12 | 85.69±5.43 | 37.09±0.97 | 64.23±1.33 | 71.21±1.45 | 84.05±4.48 | 76.69±1.56 | 90.28±0.42 | 87.57±1.33 |
| N2GON | 90.16±4.17 | 90.29±4.63 | 42.57±1.97 | 60.98±2.12 | 72.13±2.30 | 92.11±3.66 | 81.27±1.30 | 90.64±0.37 | 89.14±2.01 |

biomedical data presents distinct challenges due to its diverse representation methods. This data is often represented in multiple complex vector formats, resulting from various processing techniques. Consequently, our evaluation also encompasses these alternative representation methods.

## 5.1. Network Benchmark Datasets

**Datasets.** To evaluate the effectiveness of N2GON across a spectrum of datasets, we conducted a comprehensive analysis using 9 benchmark network datasets. These datasets are diverse, varying in domain, size, and the degree of data smoothness. Our selection encompasses three standard homogeneous citation datasets, as mentioned in (Kipf & Welling, 2017), as well as six well-known heterogeneous datasets, referenced in (Pei et al., 2020). The statistical information of these datasets are systematically outlined in Table 1. The network benchmark datasets focus on ***graph node classification***, measuring the ***accuracy*** of node classification. Details of datasets are as follows. For homophily datasets, the benchmarks include CiteSeer, PubMed, and Cora (Kipf & Welling, 2017), which are prominent in the domain of citation network analysis. In these datasets, nodes are utilized to represent academic papers, while edges correspond to the citations amongst these papers. The labels assigned to each node categorize the paper according to its research topic. As for heterophily datasets, this group encompasses datasets from the WebKB collection for the Uni-

versity of Texas, University of Wisconsin, and Cornell University, as well as Squirrel and Chameleon datasets based on Wikipedia topics, and the Actor network (Pei et al., 2020). These datas build networks through nodes (web pages or actors) and edges (hyperlinks or collaborations). The node labels are based on metrics such as webpage traffic or the categorization of actors in their context.

Given that these benchmark datasets predominantly offer processed graph data, we were tasked with adapting this data into a GON format suitable for our analysis. To achieve this, we implemented a strategy wherein the $k$-hop neighbor nodes of each node were sampled to induce a corresponding subgraph, effectively representing that particular node. This transformation method is justified by considering, for instance, a citation network where a paper's composition is not only its content but also the referenced works it cites; similarly, in social networks, a user's (node's) social connections are an integral part of their identity.

**Baselines.** We compare the proposed model N2GON with various baselines, including (1) MLP; (2) standard node-interaction GNN methods: GCN (Kipf & Welling, 2017), GAT (Veličković et al., 2018), GraphSAGE (Hamilton et al., 2017) and GCNII (Chen et al., 2020), MixHop (Abu-El-Haija et al., 2019), GCN-ChebyNet (Defferrard et al., 2016), GCN+JK (Xu et al., 2018); (3) heterophilic GNNs with adaptive node interaction: H2GCN (Zhu et al., 2020), WRGAT (Suresh et al., 2021), ACM-GCN (Luan et al.,

Table 2: Predictive performance results, ROC-AUC and PR-AUC (%), on 4 relationship classification tasks PPI, PEPMHC, TCR, and MTI. The highest performance is highlighted, and the second best performance is underlined.

| Method | PPI:HuRI | | PEPMHC:MHC1 | | TCR:Weber | | MTI:miRTarBase | |
|---|---|---|---|---|---|---|---|---|
| | ROC-AUC | PR-AUC | ROC-AUC | PR-AUC | ROC-AUC | PR-AUC | ROC-AUC | PR-AUC |
| AAC | $88.81 \pm 4.14$ | $90.62 \pm 3.09$ | $90.57 \pm 9.01$ | $83.18 \pm 4.39$ | $86.51 \pm 6.91$ | $83.97 \pm 4.20$ | $86.48 \pm 7.36$ | $81.24 \pm 5.74$ |
| ConjointTriad | $87.43 \pm 5.29$ | $91.01 \pm 7.18$ | $93.19 \pm 5.29$ | $87.21 \pm 3.52$ | $85.96 \pm 6.77$ | $83.98 \pm 8.34$ | $84.40 \pm 5.35$ | $82.82 \pm 4.56$ |
| Quasi-Seq | $79.28 \pm 4.72$ | $82.62 \pm 5.23$ | $90.75 \pm 3.19$ | $85.08 \pm 2.99$ | $76.65 \pm 4.90$ | $73.15 \pm 5.21$ | $71.44 \pm 2.66$ | $73.16 \pm 3.29$ |
| ESPF | $82.94 \pm 6.20$ | $86.38 \pm 4.85$ | $93.31 \pm 5.12$ | $87.46 \pm 6.18$ | $85.51 \pm 3.96$ | $83.32 \pm 5.69$ | $89.53 \pm 6.71$ | $85.01 \pm 4.55$ |
| CNN | $87.19 \pm 3.38$ | $90.45 \pm 6.17$ | $91.36 \pm 4.27$ | $84.99 \pm 3.86$ | $86.24 \pm 6.30$ | $83.72 \pm 3.49$ | $87.81 \pm 4.28$ | $81.39 \pm 6.50$ |
| CNN_RNN | $86.61 \pm 3.77$ | $89.81 \pm 5.10$ | $86.25 \pm 7.23$ | $81.30 \pm 4.38$ | $71.38 \pm 6.96$ | $76.11 \pm 3.32$ | $82.11 \pm 3.45$ | $80.41 \pm 3.67$ |
| Transformer | $88.20 \pm 4.91$ | $90.35 \pm 4.16$ | $91.78 \pm 4.39$ | $85.28 \pm 3.42$ | $86.30 \pm 4.76$ | $84.35 \pm 3.80$ | $88.45 \pm 6.01$ | $82.22 \pm 4.79$ |
| N2GON | $89.83 \pm 3.52$ | $91.37 \pm 5.23$ | $93.77 \pm 6.23$ | $86.94 \pm 4.65$ | $89.78 \pm 7.38$ | $89.75 \pm 2.35$ | $91.55 \pm 7.49$ | $84.66 \pm 3.95$ |

Table 3: Predictive performance results, ROC-AUC and PR-AUC (%), on DTI datasets. The best performance is highlighted, and the second best performance is underlined.

| Method | DAVIS | | KIBA | |
|---|---|---|---|---|
| | ROC-AUC | PR-AUC | ROC-AUC | PR-AUC |
| Morgan+AAC | $85.19 \pm 6.12$ | $76.77 \pm 4.63$ | $86.90 \pm 5.53$ | $83.48 \pm 6.70$ |
| Morgan+ESPF | $85.96 \pm 3.07$ | $75.93 \pm 7.58$ | $87.13 \pm 6.88$ | $85.56 \pm 3.59$ |
| Morgan+CNN | $82.01 \pm 4.78$ | $73.96 \pm 4.09$ | $90.36 \pm 8.61$ | $87.43 \pm 5.89$ |
| PubChem+AAC | $86.87 \pm 3.29$ | $77.75 \pm 7.99$ | $85.51 \pm 8.14$ | $84.74 \pm 4.98$ |
| PubChem+ESPF | $86.13 \pm 3.67$ | $78.40 \pm 4.91$ | $85.72 \pm 8.38$ | $82.23 \pm 3.52$ |
| PubChem+CNN | $88.23 \pm 5.22$ | $81.18 \pm 4.16$ | $92.61 \pm 7.36$ | $90.72 \pm 6.27$ |
| Daylight+AAC | $85.54 \pm 3.78$ | $81.79 \pm 6.18$ | $83.95 \pm 6.29$ | $76.78 \pm 5.22$ |
| Daylight+ESPF | $84.13 \pm 4.80$ | $76.85 \pm 4.34$ | $85.84 \pm 8.52$ | $82.04 \pm 6.66$ |
| Daylight+CNN | $84.43 \pm 3.32$ | $75.78 \pm 7.08$ | $91.33 \pm 6.92$ | $92.30 \pm 5.99$ |
| RDKIT+AAC | $84.71 \pm 6.94$ | $80.80 \pm 7.32$ | $87.02 \pm 6.38$ | $84.86 \pm 4.45$ |
| RDKIT+ESPF | $85.99 \pm 4.47$ | $75.19 \pm 6.52$ | $88.92 \pm 5.33$ | $82.26 \pm 4.07$ |
| RDKIT+CNN | $84.52 \pm 5.33$ | $78.56 \pm 5.71$ | $88.68 \pm 5.32$ | $85.45 \pm 6.47$ |
| ESPF+AAC | $85.38 \pm 6.52$ | $77.58 \pm 7.45$ | $87.54 \pm 7.97$ | $84.44 \pm 4.94$ |
| ESPF+ESPF | $85.14 \pm 3.04$ | $76.83 \pm 7.29$ | $85.80 \pm 5.43$ | $69.81 \pm 4.22$ |
| ESPF+CNN | $84.83 \pm 3.31$ | $77.65 \pm 6.69$ | $88.45 \pm 8.60$ | $85.14 \pm 5.72$ |
| CNN+AAC | $83.73 \pm 5.42$ | $80.21 \pm 4.57$ | $86.87 \pm 6.88$ | $85.15 \pm 6.55$ |
| CNN+ESPF | $82.96 \pm 4.70$ | $74.81 \pm 7.91$ | $85.55 \pm 5.34$ | $83.43 \pm 6.31$ |
| CNN+CNN | $84.39 \pm 3.06$ | $76.39 \pm 7.34$ | $88.46 \pm 8.06$ | $84.58 \pm 6.30$ |
| MPNN+AAC | $86.27 \pm 6.87$ | $78.96 \pm 6.56$ | $81.41 \pm 6.97$ | $83.41 \pm 5.76$ |
| MPNN+ESPF | $85.99 \pm 3.75$ | $80.93 \pm 4.26$ | $81.09 \pm 7.54$ | $78.51 \pm 6.85$ |
| MPNN+CNN | $82.78 \pm 3.86$ | $73.59 \pm 7.10$ | $89.41 \pm 5.47$ | $86.76 \pm 5.97$ |
| N2GON | $91.53 \pm 4.26$ | $83.68 \pm 5.65$ | $95.27 \pm 5.15$ | $95.08 \pm 3.11$ |

Table 4: Predictive performance results, ROC-AUC and PR-AUC (%), on DDI datasets. The best performance is highlighted, and the second best performance is underlined.

| Method | TWOSIDES | |
|---|---|---|
| | ROC-AUC | PR-AUC |
| Morgan | $82.68 \pm 5.17$ | $79.30 \pm 8.76$ |
| Pubchem | $81.69 \pm 6.40$ | $78.24 \pm 8.03$ |
| Daylight | $81.97 \pm 4.11$ | $78.76 \pm 5.59$ |
| RDKIT | $82.29 \pm 4.82$ | $78.75 \pm 6.74$ |
| CNN | $62.29 \pm 5.24$ | $79.35 \pm 7.16$ |
| CNN_RNN | $62.58 \pm 5.77$ | $78.61 \pm 5.17$ |
| Transformer | $70.30 \pm 5.18$ | $78.26 \pm 5.35$ |
| MPNN | $80.27 \pm 3.15$ | $77.34 \pm 5.70$ |
| GoGNN | $83.03 \pm 4.72$ | $79.19 \pm 5.85$ |
| N2GON | $85.53 \pm 5.67$ | $80.96 \pm 6.35$ |

2022), GGCN (Yan et al., 2022), FAGCN (Bo et al., 2021), Geom-GCN (Pei et al., 2020) (4) sampling GNNs: Fast-GCN (Chen et al., 2018), AS-GCN (Huang et al., 2018), ClusterGNN (Chiang et al., 2019), GraphSAINT (Zeng et al., 2020); and (5) decoupled GNNs: S2GC (Zhu & Koniusz, 2020), SIGN (Frasca et al., 2020), APPNP (Klicpera et al., 2019), GPRGNN (Chien et al., 2021), DAGNN (Liu et al., 2020), HopGNN (Chen et al., 2023). We report results from previous studies using the same experimental setup when available. For unreported results with available codes, we implement them using official codes.

**Implementation Details.** In this study, we utilized PyTorch to implement our methodology. Our experimental setup consisted of a server equipped with two NVIDIA A6000 GPUs running Ubuntu 20.04. We configured the hidden dimension of N2GON as 32 for the nine datasets. We tune the hops $k$ from $\{1, 2, \ldots, 6\}$. We determined the layer counts for *Encoder I* and *Encoder II*, denoted as $L_1$ and $L_2$, by selecting

from the set $\{1, 2, 3\}$. The selection of the probability parameter $\alpha$ ranged from $\{0.1, \ldots, 0.6\}$, while the parameter temperature $\tau$ was chosen from a range of $\{0.1, \ldots, 0.5\}$. For the training phase, the Adam optimizer (Kingma & Ba, 2014) was utilized. Regarding the prediction tasks, our benchmarks included a network dataset aimed at predicting graph node labels and a biomedical dataset focused on identifying relationships between graph nodes. Accordingly, our model was adapted to each dataset's unique requirements: the former directly leveraged the labels of nodes, while in the latter, treated adjacent nodes as sharing the same labels.

**Results.** Table 1 displays the node classification results comparing our N2GON algorithm with various established graph representation learning algorithms. The data in Table 1 indicates that N2GON outperforms the SOTA algorithms across a range of datasets. Notably, on the homogeneous dataset CiteSeer, N2GON shows a 3.94% improvement in performance over GCNII, and on the heterogeneous dataset ACTOR, it exceeds the performance of GGCN by 5.03%. These results suggest the potential effectiveness of N2GON in various dataset contexts. The results collectively suggest that the GON data structure, and our algorithm's approach to leveraging this structure, could offer certain advantages in capturing and representing the complexities of different types of datasets.

Table 5: The statistics of the biomedical datasets. # En1 and # En2 represent entity counts in the first and second sets, respectively, and # Link signifies the number of interactions.

| Data | DAVIS | KIBA | TWOSIDES | HuRI | MHC-1 | miRTarBase | Weber |
|---|---|---|---|---|---|---|---|
| **# En1** | 68 | 2,068 | 645 | 8,248 | 43,018 | 3,465 | 192 |
| **# En2** | 379 | 229 | 645 | 8,248 | 150 | 21,242 | 23,139 |
| **# Link** | 25,772 | 117,657 | 63,462 | 51,813 | 185,985 | 400,082 | 47,182 |

## 5.2. Biomedical Datasets

**Datasets.** We conducted an evaluation of N2GON on biomedical datasets, which includes 7 datasets from diverse domains: Drug-Target Interaction (DTI) with datasets DAVIS and KIBA (Davis et al., 2011; Tang et al., 2014), Drug-Drug Interaction (DDI) with Twosides (Tatonetti et al., 2012), Protein-Protein Interaction (PPI) with HuRI (Luck et al., 2020), Peptide-MHC Binding Prediction (PEPMHC) with MHC-I (Nielsen & Andreatta, 2016), MicroRNA-Target Interaction (MTI) with miRTarBase (Chou et al., 2018), and TCR-Epitope Binding Affinity (TCR) with Weber (Weber et al., 2021). The biomedical datasets focus on *biological entity interaction prediction*. The statistics of the biomedical datasets are shown in Table 5. These datasets were assembled by experts in the biomedical field, focusing on the prediction of affinity relationships between various entities (Koh et al., 2024). For example, the DTI dataset DAVIS includes data on drug-target affinities, while the DDI dataset TWOSIDES contains information on drug-drug affinities. This type of data is naturally suited for representation in the GON format. In this structure, individual nodes (e.g., drug molecules) are graph structures, and the inter-node relationships (e.g., interactions between drug molecules and targets) constitute a network of graphs. In our study, drug molecules are represented as graphs with nodes symbolizing atoms and edges denoting chemical bonds. For the node features of drug molecule construction, we can use the transformation methods provided by the library Therapeutics Data Commons to extract node features from SMILES sequences. Similarly, proteins (or peptides) are represented as graphs with nodes depicting amino acids and edges illustrating the chemical bonds between these amino acids.

**Baselines.** In the analysis of biomedical datasets, it's common to use one or two encoders for encoding entities like drugs or proteins. The resulting encoded vectors are then fed into a decoder to predict the relationship between two entities. Typically, this prediction task falls under binary classification, where an output of 1 signifies a relationship, and 0 denotes its absence. To establish a comprehensive baseline for comparative analysis, various encoders for drugs and proteins, specified in a domain-specific benchmark library, were employed. The drug encoding options include Morgan, Pub-

chem, Daylight, RDKIT, CNN, CNN_RNN, Transformer, and MPNN, while protein encoding options are AAC, ConjointTriad, Quasi_seq, ESPF, CNN, CNN_RNN, and Transformer. Details of these encoders can be found in (Huang et al., 2020). Therefore, our study conducts comparisons of N2GON against encoder-based classification. Additionally, since the GoGNN (Wang et al., 2021) is specifically designed for drug-drug interaction data, we also compare N2GON with GoGNN on the DDI dataset Twosides.

**Implementation Details.** For the baselines, we followed the default settings from the benchmark library (Huang et al., 2020), and the original work (Wang et al., 2021) with its specified hyperparameters. Our N2GON model, being well-suited for affinity network data, utilizes the adjacency matrix to construct the label identity matrix, assuming neighboring nodes share the same label. In these tasks, N2GON aims to predict connections between two nodes, and the aforementioned assumptions align seamlessly with the requirements of the tasks. We maintained consistent hyperparameter settings for N2GON as outlined previously. It should be noted that within these datasets, PPI and DDI are homogeneous networks, whereas other datasets, such as DTI and MTI, are heterogeneous networks, meaning the nodes have differing properties. For heterogeneous networks, we utilize the method described in the paper (Schlichtkrull et al., 2018) to convert a homogeneous GNN model into its heterogeneous equivalent. The edge splits for training, validation, and testing datasets were uniformly distributed across all methods using an 80/10/10 ratio. Note that during test phase, the encoder utilizes edges from both the training and validation sets for encoding and subsequently predicts edges from the testing set. We conduct 10 runs and report the average and standard deviation of ROC-AUC and PR-AUC scores for each method.

**Results.** Tables 2, 3, and 4 present the comparative performance results of our N2GON algorithm against corresponding baselines. Analysis of these tables reveals that N2GON consistently outperforms other methods across all datasets. For example, on the KIBA dataset within the DTI domain, N2GON shows a performance improvement of 2.66% with regard to ROC-AUC over the top encoder-based classification method, Pubchem+CNN_RNN. This suggests that N2GON may offer advantages over traditional encoder-based method and the GoGNN, in the context of biomedical data.

## 5.3. Analysis of N2GON

**Ablation Study.** In this analysis, we evaluate the impact of the constraint network components, *Grapher I* and *Grapher II*, on the N2GON model's performance. We compare N2GON's effectiveness both with and without these components, designating the models without *Grapher I* and

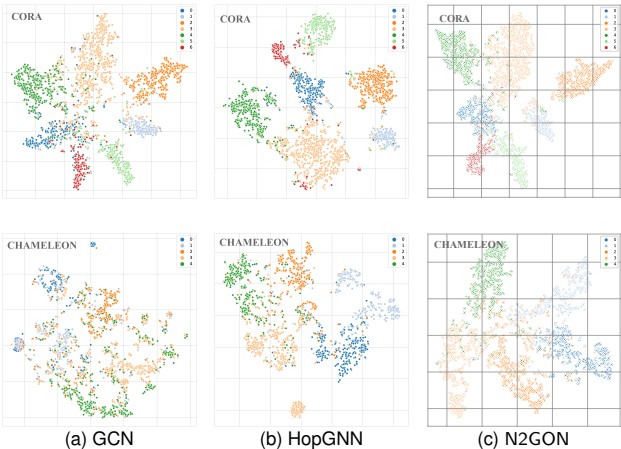

Figure 3: t-SNE embeddings of nodes in the network data.

*Grapher II* as w/o *Grapher I* and w/o *Grapher II*, respectively. The comparative performance data for these models is delineated in Table 6. An examination of the results shows that the versions of the model lacking *Grapher I* and *Grapher II* generally exhibit lower performance compared to the full implementation of N2GON. It is noteworthy that the performance of w/o *Grapher II* is superior to that of w/o *Grapher I*, indicating a greater impact of *Grapher I* on the overall effectiveness of N2GON. This finding suggests that *Grapher I* is a fundamental component for *Grapher II*'s functionality, as the construction of an importance-based constraint network relies on the presence of similar or identical labels between nodes. Thus, these results collectively indicate that the incorporation of an importance constraint network contributes positively to the performance of the proposed N2GON model.

Table 6: Ablation study on the key components of N2GON.

| Variants | Squirrel | Cora | Citeseer | Pubmed |
|---|---|---|---|---|
| N2GON | $60.98_{\pm2.12}$ | $89.14_{\pm2.01}$ | $81.27_{\pm1.30}$ | $90.64_{\pm0.37}$ |
| w/o *GrapherII* | $59.10_{\pm2.48}$ | $87.22_{\pm1.83}$ | $78.76_{\pm2.33}$ | $88.60_{\pm0.36}$ |
| w/o *GrapherI* | $58.34_{\pm3.16}$ | $85.81_{\pm1.35}$ | $77.93_{\pm1.51}$ | $88.55_{\pm0.42}$ |

**Visualization Analysis.** To demonstrate the strengths of our model in an objective manner, we utilized 2D projections for visual analysis, as shown in Fig.3. These projections depict the representations of the Cora and Chameleon datasets as processed by GCN, HopGNN, and N2GON, using the $t$-SNE algorithm (Van der Maaten & Hinton, 2008). The visualizations employ different colors to distinguish between various classes of nodes. In these representations, N2GON notably achieves a more pronounced separation, implying an enhanced ability to maintain class structures when compared to the other methods. This distinct separation achieved by N2GON suggests its effectiveness in accurately capturing

and differentiating between node classes, highlighting its potential utility in such applications.

## 6. Conclusion

In this paper, we investigate a structure, graph-of-net (GON), to model real-world systems. GON provides a multi-level perspective on the coupled dependency relations of objects, it contains: 1) node as graph, where each such graph can model entities like computing networks; 2) edge connect graph. We propose a position-aware neural network to learn the representations of GON, integrating both the structural information of nodes and their interdependencies, while also considering auxiliary information between nodes (i.e., label). To demonstrate the superiority of the GON representation and the effectiveness of our proposed model, experiments were conducted across various types of datasets. The results show that our model significantly outperforms state-of-the-art algorithms in these domains.

## 7. Limitations and Future Work

In GON, noise in any part of the data can adversely affect the learned representations. For instance, inaccuracies in node attributes or connectivity can propagate through successive encoding layers, potentially degrading overall performance. To mitigate this issue, future work may explore noise-robust strategies, including more sophisticated regularization techniques applied across multiple layers.

The process of constructing both intra-graph and inter-graph connections can influence the performance of the model. Different graph construction strategies—such as diverse node/edge selection criteria or varying ways of creating inter-graph links—could alter the quality of the resulting representations. A comprehensive comparative study on multiple construction methods would therefore be valuable. Such an investigation would offer insights into the most effective techniques for building multi-layer graphs in different applications.

In addition, several integration approaches can be explored with GON in future work. For example, integrating reinforcement learning could allow for dynamic graph adaptation, where RL agents iteratively rewire inter-graph connectivity in applications like drug discovery and evolving recommender systems. Additionally, meta-learning approaches can be used to pre-train GON encoders on diverse tasks, especially when encountering limited labeled data.

## Acknowledgments

This project was in part supported by the following projects: the National Natural Science Foundation of China (No.62432003, No.92267206).

## Impact Statement

This paper presents work whose goal is to advance the field of Machine Learning. There are many potential societal consequences of our work, none which we feel must be specifically highlighted here.

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
