# OpenReview forum: "N2GON: Neural Networks for Graph-of-Net with Position Awareness"
_ICML.cc/2025/Conference — ICML 2025 poster_

### Official Review · Reviewer_eUvt · 2025-03-07

**Overall Recommendation:** 4

**Summary:**

This paper introduces Graph-of-Net (GON), a novel graph structure where each node is itself a graph, enabling multi-level modeling of complex systems that involve hierarchical relationships. Examples include biological networks (e.g., protein-protein interactions, where individual proteins are represented as graphs within a larger network) and citation networks (where papers, modeled as text graphs, are interconnected). The authors propose N2GON, a position-aware neural network designed to learn node representations in GONs by jointly modeling intra-graph (within-node) and inter-graph (between-node) connections.

**Claims And Evidence:**

The claims made in the paper are generally supported by clear evidence.

**Essential References Not Discussed:**

n/a

**Experimental Designs Or Analyses:**

The experimental designs and analyses presented in the paper generally appear to be sound and robust, supporting the claims made about the Graph-of-Net (GoN) model.

**Methods And Evaluation Criteria:**

The methods and evaluation criteria are largely appropriate for the problem of multi-level graph learning, with strengths in architectural design and empirical breadth.

**Other Comments Or Suggestions:**

1. The explanation of the GoN concept could be further strengthened, especially with regard to how the complexity of "graphs within nodes" is handled during the modeling process. For example, it would be helpful to clarify how GoN manages the hierarchical structures within the graphs of nodes.

**Other Strengths And Weaknesses:**

**Strengths:**
1. The paper is well-written and easy to understand.
2. The proposed Graph-of-Net (GoN) structure is an innovative contribution. By representing each node as a graph, GoN enables multi-level representations of individual nodes.
3. The "position-aware neural network" mechanism introduced in the paper is a meaningful enhancement to model capabilities. This mechanism allows the model to not only focus on node features but also process interactions and dependencies between nodes.
4. The paper provides thorough experimental validation on multiple datasets, covering a range of domains such as social networks, citation networks, and biomedical networks. The experimental results are impressive.

**Weaknesses:**
1. In the introduction, the definition of Graph-of-Net (GoN) is somewhat vague. Although a general conceptual understanding is provided, a more precise mathematical definition may be needed. For instance, the definition mentions that "each node is itself a graph," but the theoretical aspects of how the "graph" size, structure, and features are defined might not be sufficiently clear. Further clarification of how GoN differs from traditional graphs, especially in terms of modeling multi-dimensional relationships in hierarchical node structures, would be helpful.
2. In some formulas (e.g., the PPR algorithm), the derivation process could benefit from more detailed explanations. For new readers, the PPR algorithm may not be immediately familiar, and providing a more comprehensive derivation would aid in understanding.

**Questions For Authors:**

Refer to the "Other Strengths and Weaknesses" part.

**Relation To Broader Scientific Literature:**

1. The key contributions of the paper, namely the introduction of the Graph-of-Net (GoN) structure and the position-aware neural network model, build upon several foundational ideas and methods from the broader graph learning and neural network literature. The paper situates itself within a well-established body of work, while introducing novel elements that extend existing techniques.
2. The idea of nodes being graphs themselves shares conceptual similarities with hypergraphs and heterogeneous graphs. In these graph structures, edges (or hyperedges) can connect more than two nodes, and the nodes can represent different types of entities or relationships. GoN extends this idea by introducing a more general notion where nodes are not just connected through edges but are, in fact, entire graphs with their own structure.

**Theoretical Claims:**

n/a

---

> ### Author Rebuttal · Authors · 2025-04-01
>
> **Q1. The Introduction's definition of Graph-of-Net (GON) is vague and needs a precise mathematical formulation of each node-as-graph (size, structure, features) and clarification of how it differs from traditional graphs in modeling hierarchical, multi-dimensional relationships.**
>
> >R1: Thank you for your insightful comments. In our framework, GON is designed as a hierarchical graph where each node is not a single data point but an entire graph. Formally, let the top-level graph (network) be defined as $\mathcal{G}^N = ( \mathcal{V}_G, \mathcal{E}_G )$, where $\mathcal{V}_G$ is the set of graph nodes and $\mathcal{E}_G$ denotes the edge set. For each node $v_G \in \mathcal{V}_G$, we associate a graph: $\mathsf{G}_v = (V, E, X)$. Here, $V$ represents the nodes within the graph, $E$ is the set of edges among these nodes, and $X$ is the feature matrix corresponding to the graph nodes. The size of each graph is determined by $|V|$, which depends on the inherent structure or the domain-specific construction of the graph. The graph structure, represented by $E$, captures the internal relationships among the nodes in the graph. This structure may vary depending on the level of detail or domain-specific insights desired. Each graph is equipped with a feature representation $X$ that encodes the characteristics of the nodes in $V$. The formation of these features can be based on raw data attributes or results from a prior processing step.
> >
> >Traditional graph models represent data as a general graph where each node corresponds to an atomic data point. In contrast, GON captures multi-dimensional relationships by explicitly modeling two levels of interaction. This dual-level representation allows GON to be particularly effective for complex hierarchical data, as it provides the capacity to model nested relationships and capture both local and global patterns within the data.
>
> **Q2. In some formulas (e.g., the PPR algorithm), the derivation process could benefit from more detailed explanations. Providing a more comprehensive derivation would aid in understanding.**
>
> >R2: The Personalized PageRank algorithm provides a way to measure how “close” or “important” one node is relative to another within a network. Imagine a random walker who starts at a specific node in the network. Instead of wandering the network entirely at random, the walker follows a rule: at each step, they decide either to move to one of the neighboring nodes or to jump back to the starting node. This jump-back mechanism ensures that the influence of the starting node remains strong throughout the walk.
> >
> >What makes PPR particularly useful is that it considers not only the direct connections between nodes but also the broader network structure. In simple terms, it captures the idea that even if two nodes share the same label or initial property, they can have varying levels of relatedness depending on how they are connected within the network.
> >
> >By adopting this method, our approach goes beyond simply saying, "nodes with the same label are similar." Instead, we are able to gauge the subtle nuances in how closely nodes are related based on both their labels and their positions within the graph structure. This allows our model to better capture complex relationships and provides a more refined way of measuring similarity among node graphs.
>
> **Q3. It would be helpful to clarify how GON manages the hierarchical structures within the graphs of nodes.**
>
> >R3: Thank you for the valuable feedback. In a Graph-of-Net (GON), each high-level node represents an entire graph that can have its own structure and detailed relationships. Instead of treating every graph as a monolithic object, we decompose the problem into two levels: 1) At the lower-level, we focus on extracting meaningful representations from the individual graphs. We utilize graph neural networks to process each graph, effectively summarizing its properties into a fixed-size embedding. 2) Once each graph has been transformed into a representation, these embeddings serve as the nodes for the higher-level graph. The interrelations among these nodes are then modeled, combining both the abstract representation of the graphs and the positional information within the larger network. The two-stage process helps us manage the inherent complexity of graphs that reside within nodes. We will add above description in subsequent revisions of the paper.

---

> > ### Comment · Reviewer_eUvt · 2025-04-02
> >
> > I appreciate the authors' detailed reply. My concerns have been largely addressed, therefore I'd like to raise my score. Please include the discussions of relevant content in the new version of this paper.

---

> > > ### Author Response · Authors · 2025-04-03
> > >
> > > We are pleased to have addressed the reviewers' concerns and are grateful for your recognition of our work. We will incorporate the above discussion into the revised version of the manuscript.

---

### Official Review · Reviewer_bLhx · 2025-03-11

**Overall Recommendation:** 3

**Summary:**

The paper introduces a novel framework called Graph-of-Net (GON), which extends traditional graph structures by modeling each node as a graph itself, creating a multi-level perspective on relationships between objects. This approach enables the capture of both the internal structure of individual nodes and the broader network of dependencies. To learn node representations within GON, the paper proposes a position-aware neural network model, which processes both intra-graph and inter-graph connections. The model incorporates dual encoders and graph constructors to build and refine a constraint network, where nodes are adaptively arranged based on their positions, as determined by the network’s constraint system.

**Claims And Evidence:**

The claims made in the paper are generally well-supported by clear and convincing evidence. The novelty of the Graph-of-Net (GON) structure is justified through the discussion and examples from general networks and biological systems. The position-aware neural network model is supported by the detailed methodology, including the dual encoders and graph constructors, which capture intra- and inter-graph relationships. The effectiveness of GON is empirically validated through extensive experiments on 16 network datasets, outperforming state-of-the-art methods.

**Essential References Not Discussed:**

N/A

**Experimental Designs Or Analyses:**

Yes, I examined the soundness and validity of the experimental design and analyses, and overall, they are largely systematic.

**Methods And Evaluation Criteria:**

Yes, the proposed methods and evaluation criteria are well-suited for the problem.

**Other Comments Or Suggestions:**

1. In the description below Equation (1) in the algorithm section, "funcitons" should be corrected to "functions."

2. The similarity function ( \text{sim}(\cdot, \cdot) ) in Equation (6) lacks a clear explanation of its basis (e.g., if using cosine similarity, the vector normalization step should be specified).

3. "node graph" → It is recommended to standardize this term as "node-graph" (with a hyphen).

4. The paper would benefit from additional discussion on the future research directions. For example, investigating how GON could be integrated with other deep learning techniques (e.g., reinforcement learning, meta-learning) to adapt to new or evolving graphs could open up interesting lines of inquiry.

**Other Strengths And Weaknesses:**

Strengths

1. The concept of treating each node as a graph is intereting. The GON structure extends traditional graph nodes into subgraphs (Graph-as-Node), enabling a shift from single-level (node-edge) modeling to multi-level modeling (subgraph internal structure + global network topology). This design is important in biological networks (such as protein-protein interactions).

2. The paper is well-written and easy to follow.

3. The experimental results are promising. The datasets are comprehensive and well-executed, covering a variety of domains including social networks, citation networks, and biomedical networks.

Weaknesses

1. The term "Position Awareness" is not strictly defined in the paper. A formal definition could be added for clarity.

2. The distinction between GON and existing hierarchical graph structures (e.g., Hierarchical Graph Networks, Hypergraphs) is not clearly quantified.

**Questions For Authors:**

See Weaknesses.

**Relation To Broader Scientific Literature:**

The paper successfully bridges gaps in hierarchical graph representation learning. Its innovations—particularly the GON structure and PPR constraints—address challenges in modeling multi-scale systems, positioning it as a meaningful contribution to the broader graph learning literature.

**Theoretical Claims:**

N/A

---

> ### Author Rebuttal · Authors · 2025-04-01
>
> **Q1. The term "Position Awareness" is not strictly defined in the paper. A formal definition could be added for clarity.**
>
> >R1:  Thank you for highlighting the need for a more formal definition of the term "Position Awareness." In our work, we use "Position Awareness" to refer to the model’s ability to capture and integrate the relative placement and connectivity of nodes within the overall network structure. Position Awareness is the property of a network model whereby each node's representation explicitly encodes its structural context within the graph. This encoding captures not only the node’s intrinsic features (e.g., labels or attributes) but also its topological characteristics—such as how it is connected to other nodes, and its overall influence in the network. In our framework, this is operationalized by leveraging a scoring mechanism (i.e., via the Personalized PageRank algorithm) that quantifies the relative importance or influence of nodes.
> >
> >By assigning a score that measures the likelihood of reaching one node from another through random walks with restarts, our approach formalizes a node’s “position” within the network. This score effectively differentiates nodes that might share similar labels but occupy distinct roles depending on their connectivity pattern and centrality in the graph. These scores are then integrated into the node representations, ensuring that position-dependent information contributes to the final embedding. As a result, the model is better equipped to capture nuanced relationships that go beyond simple label similarity.
> >
> >We will include this formal definition in the revised manuscript to enhance the clarity and rigor of our contribution.
>
> **Q2. The distinction between GON and existing hierarchical graph structures (e.g., Hierarchical Graph Networks, Hypergraphs) is not clearly quantified.**
>
> >R2: Thank you for your comment. We distinguish our Graph-of-Nets (GON) framework from other hierarchical graph structures as follows: 1) **Explicit Two-Level Representation:** In GON, each node is a complete graph that retains its full internal structure. This is different from many hierarchical graph networks, which typically process standard graphs where the nodes are represented as vectors rather than graphs, potentially losing fine-grained structural details. 2) **Preservation of Intra-Node Complexity:**  Instead of merging all details into a single representation, GON processes each graph independently (using specialized graph neural networks) and then integrates these detailed embeddings into the higher-level network. This two-stage approach effectively captures both local (intra-node) and global (inter-node) relationships.
>
> We will add these distinctions in the revised manuscript.
>
> **Q3. In the description below Equation (1) in the algorithm section, "funcitons" should be corrected to "functions."**
>
> >R3: Thank you for catching the typo. We appreciate your attention to detail, and we will correct in the revised manuscript.
>
> **Q4. The similarity function ( \text{sim}(\cdot, \cdot) ) in Equation (6) lacks a clear explanation of its basis (e.g., if using cosine similarity, the vector normalization step should be specified).**
>
> >R4: Thank you for your comment. We clarify that in Equation (6), we directly compute the cosine similarity on the final representations without any additional normalization. We will update the manuscript to clearly state this.
>
> **Q5. "node graph" → It is recommended to standardize this term as "node-graph" (with a hyphen).**
>
> >R5: Thank you for your comment. We appreciate your suggestion to standardize the term. We will update the manuscript to use "node-graph" consistently.
>
> **Q6. The paper would benefit from additional discussion on the future research directions. For example, investigating how GON could be integrated with other deep learning techniques (e.g., reinforcement learning, meta-learning) to adapt to new or evolving graphs could open up interesting lines of inquiry.**
>
> >R6: Thank you for the insightful suggestion. We plan to explore several integration approaches with GON in our future work. For example, integrating reinforcement learning could allow for dynamic graph adaptation, where RL agents iteratively rewire inter-graph connectivity in applications like drug discovery and evolving recommender systems. Additionally, we are considering meta-learning approaches to pre-train GON encoders on diverse tasks—enabling rapid adaptation to new scenarios, especially when encountering limited labeled data. Moreover, we can extend GONs to handle temporal dynamics by incorporating architectures for gradually updating both intra- and inter-graph connections in evolving systems. We believe these strategies will further enhance GON’s flexibility and robustness across various challenging, real-world applications.

---

### Official Review · Reviewer_99JV · 2025-03-12

**Overall Recommendation:** 4

**Summary:**

The paper N2GON presents a new approach to graph learning, with a focus on the Graph-of-Net structure and a position-aware neural network model. The comprehensive experimental evaluation and detailed methodology are significant strengths. However, the paper could be further improved by including runtime comparisons, expanding the related work section, and correcting some grammatical errors.

**Claims And Evidence:**

Yes, the claims made in the paper are well-supported by the evidence.

**Essential References Not Discussed:**

N/A.

**Experimental Designs Or Analyses:**

The experimental designs and analyses in the paper are sound.

**Methods And Evaluation Criteria:**

Yes, the proposed methods and evaluation criteria align with the problem's core challenges.

**Other Comments Or Suggestions:**

1. It would be helpful to expand the discussion on real-world applications, particularly in complex domains like drug discovery, bioinformatics, or social networks. While the paper covers some biomedical datasets, offering a broader perspective on potential use cases (including possible limitations in these domains) would give readers a more complete understanding of the impact of GON.

2. There are some grammatical errors and typos in the paper:

Line 23: "an" should be changed to "a".
Line 60 (left column): "applicable" should be removed.
Line 123 (right column): "are" should be "is".
Line 155 (right column): "possess" should be in plural form.
Line 267 (left column): "usally" should be corrected to "usually".
Line 261 (right column): "is" should be "are".

**Other Strengths And Weaknesses:**

Pros
1. I appreciate the motivation behind this paper. The concept of representing each node as a graph within a larger network is innovative and extends traditional graph structures.
2. The proposed algorithm integrates both intra-graph and inter-graph connections, and incorporating PPR to capture the relative position of node graphs is interesting.
3. The experimental results also seem to validate the effectiveness of the proposed model.

Cons
1. Although the paper provides a complexity analysis in the appendix, it lacks experimental comparisons in terms of runtime performance. Including an intuitive comparison of computational efficiency could enhance the paper.
2. The related work section is not comprehensive enough. More discussion on position-awareness in graph learning should be included.
3. In the experimental section, although multiple datasets are mentioned, the paper does not provide sufficient details on their specific characteristics (e.g., the number of nodes, edges, and class distribution).

**Questions For Authors:**

In addition to the weaknesses mentioned earlier, I have the following specific questions:

> 1. In Algorithm 1, the phrase "Sample all node graphs" seems contradictory. "Sample all" suggests sampling the entire set, whereas "Sample mini-batch" would indicate a subset. Could you clarify whether full-batch training is used or update the terminology accordingly?
> 2. The paper does not specify the weighting coefficient between the constraint loss \( L_{\text{con}} \) and the NLL loss. How is this coefficient set and adjusted in the algorithm?

**Relation To Broader Scientific Literature:**

The key contribution of this paper extends traditional graph models by representing each node as a graph itself, enabling a more sophisticated multi-level representation GON. It builds on prior work in graph, which have been used to model complex relationships in social and biological networks. However, existing graph models do not fully capture the hierarchical dependencies present in real-world systems. GON provides a more flexible and general framework, making it a valuable work in graph representation learning.

**Theoretical Claims:**

N/A.

---

> ### Author Rebuttal · Authors · 2025-04-01
>
> **Q1. It is recommended to add runtime comparisons.**
>
> >R1: Thank you for your comment. We conducted the runtime experiments and the results (the average elapsed time per epoch), summarized in the table below, indicate that on benchmark graphs, our runtime is comparable with that of SOTA baselines, while on biomedical datasets our method is significantly more efficient than traditional algorithms.
>
> *Table I. Running Time per epoch on Benchmark Graph Datasets (in seconds)*
>
> | Algorithms | Cora  | Ciesear | PubMed | Cornell | Texas | Actor | Squirrel | Chameleon |
> | :---: | :---: | :-----: | :----: | :-----: | :---: | :---: | :------: | :-------: |
> |    Ours    | 0.055 |  0.062  | 0.239  |  0.039  | 0.043 | 0.09  |  0.071   |   0.055   |
> |   H2GCN    | 0.005 |  0.005   | 0.016  |  0.005  | 0.004 | 0.011 |  0.088   |   0.043   |
> |   DAGNN    | 0.012 |  0.013  |  0.01  |  0.011  | 0.019 | 0.01  |  0.012   |   0.013   |
> |   HopGNN   | 0.01  |  0.01   | 0.024  |  0.012  | 0.009 | 0.013 |  0.013   |   0.012   |
>
> *Table II. Running Time per epoch on Biological Datasets (in seconds)*
>
> |   Algorithms   | PEP-MHC |  PPI   |  TCR  |   MTI   |
> | :------------: | :-----: | :----: | :---: | :-----: |
> |      Ours      |  2.00   | 0.502  | 1.17  |  1.15   |
> | ConjointTraid |  82.08  | 31.96  | 6.01  | 175.96  |
> |   Quasi-Seq    | 117.15  | 30.91  | 5.97  | 174.99  |
> |      ESPF      | 131.97  | 25.99  | 7.02  | 177.98  |
> |      CNN       | 1023.49 | 310.96 | 76.96 | 1103.97 |
> |  Transformer   | 351.97  | 88.49  | 25.29 | 1040.97 |
>
>
> **Q2. More discussion on position-awareness in graph learning should be included.**
>
> >R2: Thank you for the feedback. In our work, *position-awareness* is achieved by leveraging the Personalized PageRank algorithm, which computes the topological influence of node graphs on each other, implicitly revealing their *relative positions* in the network. In essence, once all pairwise similarities are determined, each node inherently carries information about its relative position within the similarity network structure.
>
> **Q3. Providing dataset details (e.g., node/edge counts) for clarity.**
>
> >R3: Thank you for your valuable suggestion. We have now added detailed descriptions of benchmark dataset  in the table below.
>
> |   Datasets   | **Texas** | **Wisconsin** | **Actor** | **Squirrel** | **Chameleon** | **Cornell** | **Citeseer** | **Pubmed** | **Cora** |
> | :----------: | :-------: | :-----------: | :-------: | :----------: | :-----------: | :---------: | :----------: | :--------: | :------: |
> |  **#Nodes**  |    183    |      251      |   7,600   |    5,201     |     2,277     |     183     |    3,327     |   19,717   |  2,708   |
> |  **#Edges**  |    295    |      466      |  26,752   |   198,493    |    31,421     |     280     |    4,676     |   44,327   |  5,278   |
> | **#Classes** |     5     |       5       |     5     |      5       |       5       |      5      |      7       |     3      |    6     |
>
>
> **Q4. **Discuss more real-world applications (e.g., drug discovery) and potential limitations to better illustrate GON's impact.**
>
> >R4: Thank you for your valuable suggestions. We will further expand the discussion of GON’s practical applications in the manuscript. For example, in drug discovery, GON can model drug molecules as graphs of atoms while representing proteins as residue graphs, capturing the binding patterns between them through a global network. In bioinformatics, GON enables multi-scale analysis of protein interaction networks, such as identifying functional modules at the residue level. For social networks, GON can model user communities (e.g., user-centric social subgraphs) and the relationships across communities, though real-world deployment must address privacy concerns and data sparsity issues. Common challenges include the cost of data construction (e.g., the need for expert annotations for molecular graphs) and computational overhead.
>
> **Q5. There are some grammatical errors and typos.**
>
> >R5: Thank you for your detailed feedback. We will review the manuscript and corrected all the mentioned grammatical errors and typos.
>
> **Q6. Could you clarify if Algorithm 1 uses full-batch training or mini-batch sampling?**
>
> >R6: Thank you for your valuable feedback. For the GON datasets, we use full-batch training—processing the entire data at once—since it is feasible to handle these datasets in one go. In the revised manuscript, we will clarify this point.
>
> **Q7. How is this coefficient set for the loss in the algorithm?**
>
> >R7: Thank you for raising this important point. In our work, we deliberately did not introduce a weighting coefficient between the constraint loss ($L_{\text{con}}$​) and the NLL loss. We found that both loss components naturally operate on comparable scales, allowing us to simply sum them without additional tuning. This design choice not only simplifies the loss function but also reduces the number of hyperparameters, streamlining the training process.

---

> > ### Comment · Reviewer_99JV · 2025-04-05
> >
> > Thank you for the detailed response. The responds have largely addressed my concerns regarding the aforementioned issues. Additionally, concerning the Biomedical Datasets, I would like to better understand how the node features in the constructed graph were derived. Although Section 4.2 provides a general explanation, it does not specifically clarify how the atomic-level feature vectors are generated for drug molecules—are they based on SMILES sequences, for example? Providing a more detailed description of this process would enhance the clarity of the paper.

---

> > > ### Author Response · Authors · 2025-04-05
> > >
> > > We are pleased to have addressed most of the reviewers’ concerns and appreciate the recognition of our work. For the node features of drug molecule construction, we can use the transformation methods provided by the biomedical domain library Therapeutics Data Commons (tdcommons.ai) to extract node features from SMILES sequences. For example, each node (atom) feature is composed of five concatenated parts: a one-hot encoding representing the atomic symbol (type), a one-hot encoding representing the number of bonds connected to the atom (degree), a one-hot encoding representing the atom’s formal charge, a one-hot encoding representing the chiral tag, and a binary feature indicating whether the atom is aromatic. We thank the reviewers for your constructive suggestions and will include the above discussion in the paper.

---

### Official Review · Reviewer_afvE · 2025-03-13

**Overall Recommendation:** 3

**Summary:**

This paper introduces Graph-of-Net (GON), a novel graph structure where each node itself is a graph, enabling multi-level representation and analysis of complex real-world systems. To effectively learn representations within GONs, the authors propose N2GON, a position-aware neural network that captures both intra-graph and inter-graph interactions using dual graph encoders and an implicit constraint network. Extensive experiments demonstrate that N2GON outperforms state-of-the-art models in graph learning tasks.

**Claims And Evidence:**

Yes

**Essential References Not Discussed:**

None

**Experimental Designs Or Analyses:**

In the first part of the experiments, the partitioning of GON does not seem entirely reasonable, whereas in the later part, the partitioning in the chemical datasets appears to be more appropriate.

**Methods And Evaluation Criteria:**

Yes. However, the proposed method is designed for GON, but the way this structure is partitioned in the paper does not seem entirely reasonable. This issue is particularly evident in datasets such as CiteSeer, where the partitioning is performed directly using KNN, resulting in computed outcomes that merely replicate the information propagation process of GNN.

**Other Comments Or Suggestions:**

The full name of GON is mentioned twice in the paper.

**Other Strengths And Weaknesses:**

The paper investigates GON and proposes a more generalizable approach. However, the experimental setup does not seem sufficiently comprehensive. I believe the authors provided excellent examples of GON in the introduction, but it is unclear whether experiments on related datasets are feasible. Of course, obtaining relevant data may be challenging.

**Questions For Authors:**

None

**Relation To Broader Scientific Literature:**

Yes

**Theoretical Claims:**

N/A

---

> ### Author Rebuttal · Authors · 2025-04-01
>
> **Q1. The partitioning of GON —particularly in datasets like CiteSeer—appears questionable, where the partitioning is performed directly using KNN, resulting in computed outcomes that merely replicate the information propagation process of GNN.**
>
> >R1: Thank you for your feedback. Below, we provide clarification as follows:
>
> >1. **Rationale for $k$-hop Sampling Strategy**. We would like to clarify that we adopt a $k$-hop sampling (instead of KNN, which only involves 1-hop) to construct a graph for each node. This method is motivated by the observation that in many real-world scenarios, a node’s full identity is not encapsulated solely by its features but also by the structural context provided by its neighboring nodes.
>
> >For example, 1) in **Citation Networks** such as CiteSeer, a paper’s identity is defined not only by its content but also by its *references* and *cited-by* relationships. By sampling $k$-hop neighbors, we explicitly model a node’s "extended identity," which includes its local academic context (e.g., related works). This aligns with the GON philosophy of representing nodes as hierarchical structures. 2) in **Social Networks**, a user’s social graph (friends, followers) is a natural extension of their identity, and $k$-hop sampling preserves this contextual information. Therefore, sampling the $k$-hop neighborhood effectively captures the local context.
>
> >The reviewer noted that, in the case of CiteSeer, direct partitioning by $k$-hop might induce outcomes very similar to standard GNN information propagation. We believe this observation is, in fact, **supportive** of our methodology. The successful aggregation of neighbor information by GNNs substantiates that a node’s neighborhood plays a crucial role in the learning process. Our method does not merely replicate the GNN propagation process but rather formalizes the node’s expanded representation through explicit graph construction. This ensures that our Graph-of-Net structure inherently reflects the multi-level composition of nodes—each node graph is a manifestation of both its own features and the collective representation of its neighbors.
>
> >2. **Alternative Graph Construction Method**. Beyond $k$-hop sampling, we explored constructing graphs from raw node data. We appreciate the reviewer's note on the challenge of accessing original data. In response, we made significant efforts and successfully obtained the raw textual data for the CiteSeer data through the GitHub project (https://github.com/sivaramanl/Information-Retrieval/tree/master/Text%20Processing/citeseer). This repository provided us with nearly all the title and abstract information for each paper. For this alternative graph construction, we processed each paper’s text as follows: 1) **Text Preprocessing:** We used NLTK to split the text of each paper into sentences, treating each sentence as a node in the paper’s graph.  2) **Feature Extraction:** We generated embeddings for each sentence using the popular sentence transformer model `all-MiniLM-L6-v2`, which provided the node attributes. For papers where the original information is missing, we default to representing the paper as a single node with an all-zero attribute vector. 3) **Edge Construction:** Edges were formed by computing the cosine similarity between sentence embeddings and applying a threshold of 0.7 to retain only strong connections.
>
> >We then conducted experiments on this newly constructed data. The generated data statistics and performance comparisons, as presented in the tables below. The comparable performance of $k$-hop and text-based GONs (**81.27% vs. 81.82%**) demonstrates that both methods validly capture hierarchical semantics. The $k$-hop approach is a **pragmatic and effective** proxy when raw data is unavailable, while text-based construction confirms GON($k$-hop)’s flexibility for domains with explicit substructures.
>
> *Table I: Resulting GON Statistics on Citeseer*
>
> | Metric                      | Value       |
> |-----------------------------|-------------|
> | Avg. Nodes per Graph         | 7.72        |
> | Max Nodes                    | 336         |
> | Min Nodes                    | 1           |
> | Avg. Edges per Graph         | 156.36      |
> | Max Edges                    | 106,286     |
> | Min Edges                    | 0           |
> | Node Feature Dimension       | 384         |
>
> *Table II: Accuracy on data CiteSeer*
>
> | Algorithm              | Acc (%)     |
> | ---------------------- | ---------------- |
> | APPNP                  | 77.06 ± 1.73     |
> | GPRGNN                 | 75.56 ± 1.62     |
> | MixHop                 | 76.26 ± 1.33     |
> | FAGCN                  | 74.86 ± 2.42     |
> | DAGNN                  | 76.44 ± 1.97     |
> | HopGNN                 | 76.69 ± 1.56     |
> | **N2GON($k$-hop)**     | **81.27 ± 1.30** |
> | **N2GON (Text-Based)** | **81.82 ± 1.46** |
>
> **Q2. The full name of GON is mentioned twice.**
>
> >R2: Thank you for pointing that out. We will remove the extra name.

---

### Decision · Program_Chairs · 2025-05-01

**Decision:**

Accept (poster)

**Comment:**

This paper introduces ​​N2GON​​, a novel neural network framework designed for ​​Graph-of-Net (GON)​​, a hierarchical graph structure where each node is itself a graph. The proposed model leverages ​​position-aware learning​​ to capture both intra-graph (within-node) and inter-graph (between-node) relationships, integrating node labels and topological constraints via dual encoders and a refined constraint network. The authors demonstrate the effectiveness of N2GON through extensive experiments on ​​16 datasets​​ spanning social networks, citation networks, and biomedical applications, showing significant improvements over state-of-the-art baselines.

The paper received generally positive reviews, with reviewers acknowledging its ​​novelty​​, ​​theoretical grounding​​, and ​​empirical effectiveness​​. While initial concerns were raised, all four reviewers expressd that the responses have addressed their concerns, and consider this paper worthy of acceptance.